# Low Hysteresis and Fatigue-Resistant Polyvinyl Alcohol/Activated Charcoal Hydrogel Strain Sensor for Long-Term Stable Plant Growth Monitoring

**DOI:** 10.3390/polym15010090

**Published:** 2022-12-26

**Authors:** Lina Wang, Zhilin Zhang, Jie Cao, Wenqian Zheng, Qi Zhao, Wenna Chen, Xinye Xu, Xiaoyu Luo, Qi Liu, Ximei Liu, Jingkun Xu, Baoyang Lu

**Affiliations:** 1Jiangxi Key Lab of Flexible Electronics, Flexible Electronics Innovation Institute, Jiangxi Science and Technology Normal University, Nanchang 330013, China; 2School of Pharmacy, Jiangxi Science and Technology Normal University, Nanchang 330013, China; 3School of Chemistry and Molecular Engineering, Qingdao University of Science and Technology, Qingdao 266042, China

**Keywords:** conductive hydrogel, strain sensor, low hysteresis, fatigue-resistant, long-term stability, plant growth monitoring

## Abstract

Flexible strain sensor as a measurement tool plays a significant role in agricultural development by long-term stable monitoring of the dynamic progress of plant growth. However, existing strain sensors still suffer from severe drawbacks, such as large hysteresis, insufficient fatigue resistance, and inferior stability, limiting their broad applications in the long-term monitoring of plant growth. Herein, we fabricate a novel conductive hydrogel strain sensor which is achieved through uniformly dispersing the conductive activated charcoal (AC) in high-viscosity polyvinyl alcohol (PVA) solution forming a continuous conductive network and simple preparation by freezing-thawing. The as-prepared strain sensor demonstrates low hysteresis (<1.5%), fatigue resistance (fatigue threshold of 40.87 J m^−2^), and long-term sensing stability upon mechanical cycling. We further exhibit the integration and application of PVA-AC strain sensor to monitor the growth of plants for 14 days. This work may offer an effective strategy for monitoring plant growth with conductive hydrogel strain sensor, facilitating the advancement of agriculture.

## 1. Introduction

Understanding the influence of biological/environmental factors on plant growth is crucial [1]. Monitoring plant growth enables farmers to adjust the plant-growing environment, which is important for improving the growth and survival rate of crop production. However, many devices for monitoring plant growth are rigid, inaccurate, and expensive, immensely limiting their application in agricultural production [2,3,4,5]. Moreover, their sizes and dimensions are manually adjusted during the measurement process and are not suitable for long-term monitoring of plant growth. Therefore, it is necessary to develop flexible strain sensors to measure plant growth in real-time.

Currently, flexible strain sensors fabricated from soft materials and advanced technologies are of great significance for aerospace, robots, actuators, electronic skins, and motion detection [6,7,8,9,10]. Especially, flexible strain sensors have been used as precise measuring equipment for continuously monitoring plant growth by converting mechanical deformation into electrical resistance [11,12,13,14,15,16]. Many researches have designed stretchable strain sensors with soft materials (gelatin, hydrogels, silks, etc.) [17,18,19], but these sensors still suffer from severe limitations, such as large hysteresis, insufficient anti-resistant ability, and inferior long-term stability. These problems lead to the incapability of strain sensors for subtle change detection and long-stable stability in monitoring plant growth. Thus, rational design and fabrication of conductive hydrogel strain sensors with low hysteresis, fatigue resistance, and long-term stable monitoring of plant growth are expected to promote digitalization, precision, and smart farming [20,21].

In this work, we prepare a low hysteresis and fatigue-resistant conductive hydrogel strain sensor for long-term stable monitoring of plant growth. We choose activated charcoal (AC) as the electrical phase and polyvinyl alcohol (PVA) as the mechanical phase to prepare conductive hydrogels strain sensor. PVA is a high molecular weight polymer, which becomes a viscous solution after dissolution and can effectively prevent the sedimentation of AC particles. It is conducive to the uniform dispersion of AC particles in the PVA viscous solution to form a continuous and stable conductive network, which is beneficial for strain sensors to maintain stable performance. We demonstrate that our PVA-AC conductive hydrogel strain sensor displays low hysteresis (<1.5%) and fatigue resistance (40.87 J·m^−2^). Furthermore, our PVA-AC conductive strain sensor also achieves other excellent properties, including high stretchability (>300%), low Young’s modulus (10~30 kPa), and high linearity (~0.98). These favorable properties demonstrate that the PVA-AC conductive hydrogel strain sensor can respond to stimuli in plant growth and realize long-term stable detection of the plant growth status. This strain sensor can be an ideal candidate to satisfy the need for precise data on plant growth, providing a promising window for the development of intelligent agriculture [22,23,24,25].

## 2. Materials and Methods

### 2.1. Materials

Polyvinyl alcohol (PVA-124, Mw: 105,000) was purchased from Aladdin (Shanghai, China), and activated charcoal (surface areas: 300–2000 m^2^/g, particle size: ~1800 nm) was obtained from Nature’s Way Products (USA). VHB double-sided tape (4905, 0.5 mm) was acquired from 3M company (Shanghai, China).

### 2.2. Fabrication of Strain Sensor

*Preparation of PVA-AC Conductive Hydrogel.* PVA aqueous solution was prepared by dissolving PVA-124 powders into deionized (DI) water at 90 °C. AC particles were dispersed into PVA aqueous solution (activated charcoal to PVA aqueous solution ratio of 1 wt.%, 5 wt.%, 10 wt.%, 20 wt.%, and 30 wt.%, respectively) to prepare the PVA-AC precursor solution. Then, the PVA-AC precursor solution was poured into molds for multiple freezing-thawing (freezing at −20 °C for 8 h and thawing at 25 °C for 3 h) to prepare the PVA-AC conductive hydrogel.

*Assembly of PVA-AC Conductive Hydrogel Strain Sensor.* As-prepared PVA-AC conductive hydrogel strain sensor was a simple device with a three-layer structure, in which the VHB viscoelastic tape was used as the substrate and the encapsulation layer, the PVA-AC conductive hydrogel was treated as the electrode, and the conductive carbon clothes were explored as electrode wires to conduct electrical signals which were connected with the PVA-AC conductive hydrogel. In order to avoid inaccurate measurement results caused by water loss from the PVA-AC conductive hydrogel, we used VHB double-sided tape to encapsulate the PVA-AC conductive hydrogel. VHB elastic tape can be closely fitted with the PVA-AC conductive hydrogel. 

### 2.3. Characterization

SEM images of cross-sectional morphologies of the PVA-AC conductive hydrogel were observed by a Scanning Electron Microscope (Hitachi Regulus 8100). FTIR spectra of the PVA, AC, and PVA-AC conductive hydrogels were performed in the wavelength range of 4500–500 cm^−1^ by employing an FTIR spectrometer (Spectrum Two, Perkin-Elmer, Waltham, MA, USA). The rheological properties of the PVA-AC inks were tested by using a rotational rheometer (Discovery HR-2, TA Instruments Inc., New Castle, DE, USA).

*Mechanical Characterization.* Mechanical performances of the PVA-AC conductive hydrogels in a dumbbell shape were tested by a micro-controlled electronic universal testing machine (DWD-010 with 100 N load cell, Changchunkexin Precision Instrument). The prepared dumbbell-shaped conducting hydrogels, with a dimension of 20 × 3 × 2 mm, were immobilized in the fixture at 50 mm/min for the mechanical test.

*Fatigue-Resistant Performance of PVA-AC Conductive Hydrogel.* We utilized a single-cut method to quantify the fatigue thresholds of the PVA-AC conductive hydrogel, while the fatigue experiment was performed in a water bath for the tensile experiments to avoid water loss. We used a rectangular PVA-AC conductive hydrogel with a length of 21 mm, a thickness of 2 mm, and a width of 13 mm. First, an opening was cut in the middle of the sample (with a length of less than 1/5 of the overall length) and pre-stretched in a water bath to determine the $\lambda_{max}$ strain by adjusting the strain range. Then, we performed a fatigue experiment on the samples at a fixed $\lambda_{max}$ strain, and we recorded the crack length in the undeformed state with a camera and then recorded the crack length of the samples after 15,000 and 30,000 turns of stretching, respectively. The energy dissipation equation was calculated by $G\;\left(\lambda_{\max},\;N\right)=2k\;\left(\lambda_{\max}\right)\;\times \;c\left(N\right)\;\times \;W\;\left(\lambda_{\max},\;N\right),$ $k=3/\sqrt{\lambda_{\max}}$, where *k* is a slowly varying function of the applied stretch. By varying the stretching of $\lambda_{\max}$, we obtain a stretching curve of dc/dN per cycle versus the applied energy release rate *G*, and the fatigue threshold is obtained from the dc/dN curve.

*Sensing Characterization.* The conductive hydrogel strain sensor was fixed to the underwater stretching machine with clamps to keep it horizontal, and the electrode clamps of the Precision LCR Meter (Tonghui, TH2829C, 20 Hz~1 MHz) hold the conductive carbon cloth at both ends of the sensor. The hysteresis was calculated by H=(SL-SU )/SL, where *SL* and *SU* are the loaded area and unloaded area of a specific stress–strain curve, respectively. The calculation formula of *GF* was calculated by $\mathrm{GF}=\left(\Delta R/R_{0}\right)/\varepsilon ,$ Δ*R* and *R*_0_ is relative resistance and initial resistance, respectively. The long-term stability of the PVA-AC hydrogel sensor showed the relative resistance tested by the LCR meter with 10,000 cycles of stretching at 25 °C. Before testing the long-term cycling experiments, the PVA-AC conductive hydrogel strain sensor was pre-stretched for 100 cycles to maintain stable electrical properties.

## 3. Results and Discussion

### 3.1. Design of PVA-AC Conductive Hydrogel

Based on the requirement of strain sensors, we design conductive hydrogels as sensing layers for strain sensors using structurally stable, low-cost PVA and AC. The PVA serves as the mechanical network’s backbone for the strain sensors, and the AC is simultaneously diffused in the soft PVA substrate as a conductive filler to provide it with electrical qualities based on its mechanical properties. The PVA-AC conductive hydrogels are prepared by introducing conductive AC into the PVA viscous solution (Figure 1a). Our synthesis method involves mixing the AC particles and the PVA viscous solutions under vigorous mechanical stirring, followed by physical cross-linking via continuous freeze-thawing to form stable conductive hydrogels (Figure 1a and Appendix A).

The conductive AC is uniformly dispersed in the viscous PVA solution to form a continuous PVA-AC conductive network. The AC particles settle by gravity in the solution, and the viscosity of the PVA solution prevents the settling rate of the AC particles until the AC particles maintain equilibrium. The rheological characterization of the PVA-AC conductive hydrogel inks demonstrates that the viscous PVA solution facilitates the conductive AC to disperse (Figure 1b) [26,27,28]. The continuous conductive network eliminates the tunneling effect between the conductive particles and the substrate material, effectively minimizing hysteresis during the stretching process. The PVA-AC conductive hydrogel is formed from crystalline domains of polyhydroxylated PVA cross-linked by hydrogen bonding that provides sacrificial bonds for energy dissipation. When hydrogen bonds are broken, the PVA crystalline domain is disrupted, and the discontinuous conductive network impairs the mechanical and electrical phases of the PVA-AC conductive hydrogel. To illustrate the stability of the PVA-AC conductive hydrogel, we further elaborate on its stability in terms of the chemical structure and microstructure. The FT-IR shows that the absorption peak of the PVA-AC conductive hydrogel hardly shifted after adding the activated charcoal, which proves the stability of the PVA-AC conductive hydrogel chemical structure (Figure 1c) [29]. Furthermore, we provide scanning electron microscopy (SEM) images of the PVA-AC conductive hydrogel to demonstrate that the AC particles uniformly disperse in PVA viscous solutions. Figure 1d shows that the PVA-AC conductive hydrogel exhibits a closely arranged honeycomb structure, which can provide excellent mechanical network. Meanwhile, the homogeneous distribution of the Al element seen in the EDS images demonstrate the uniform dispersion of AC in the PVA viscous solution (Appendix A). The conductive hydrogel, composed of conductive fillers and synthetic polymers, displays rewarding flexibility and ductility after stretching, twisting, and bending (Figure 1e–g). This work demonstrates that the formation of a continuous PVA-AC conductive network is beneficial for obtaining interesting performances, such as low hysteresis, fatigue resistance, and long-term stability. Furthermore, the homogeneous dispersion of the conductive AC ensures the stable electrical properties of the PVA conductive hydrogel and achieves a stable output of the sensing signal.

### 3.2. Mechanical Properties of PVA-AC Conductive Hydrogel

The PVA-AC conductive hydrogel exhibits high stretchability and ductility and possesses a low Young’s modulus, similar to that of human tissue [30,31]. Based on its stretchable properties, we evaluate the mechanical properties of the PVA-AC conductive hydrogel. The stress–strain curves of the different proportions of PVA-AC conductive hydrogels indicate superior stretchability (Figure 2a). It can be seen that the content of AC has a great influence on the strength of the PVA-AC conductive hydrogel. The results show that as the concentration of AC increases, the strain of the hydrogels gradually decreases, and the stress gradually increases. Figure 2b and c show a monotonic increasing trend in Young’s modulus and toughness with an increasing AC concentration. The PVA-AC conductive hydrogel toughness is calculated by the integral area of the stress–strain, and the strain gradually decreased with an increasing AC content, but the stress is gradually increasing. In particular, the stress of the PVA-AC conducting hydrogel over 400 kPa at a 30 wt.% PVA-AC mass ratio, which is about 4 times higher than that of pure PVA hydrogel, but the elongation at break value is only 272% (Figure 2d), which is about 1.5 times lower than that of pure PVA hydrogel. In the sensitivity test of different concentrations of PVA-AC conductive hydrogel, the sensitivity value of 5 wt. % PVA-AC was the highest, indicating that it has the best sensing performance (Appendix A); therefore, we choose the 5 wt.% PVA-AC as the optimum ratio for our material. These results indicate that the mass ratio of the AC content has a significant effect on the mechanical properties of the hydrogel [32,33].

In addition to high stretchability, low Young’s modulus, and high toughness, the PVA-AC conductive hydrogel also has excellent stability. After undergoing 100 cycles of stretching and shrinking at 100% strain, the PVA-AC conducting hydrogel can still return to its original shape without a significant change, further evaluating the mechanics of the PVA-AC conductive hydrogel’s stability (Figure 2e). Meantime, the energy dissipation efficiency of the PVA-AC hydrogel is different under different strains and a synchronized increase between them is maintained (Figure 2f). Therefore, when the strain reaches a certain level, it will cause the PVA-AC conductive hydrogel to fracture. Overall, the PVA-AC conductive hydrogel exhibits excellent mechanical properties; it can be better attached to plants and long-term monitor the size changes during plant growth.

### 3.3. Fatigue Fracture Properties of PVA-AC Conductive Hydrogel

To further illustrate the ability of the PVA-AC conductive hydrogels to withstand long-term load cycles, we design a single-notch tensile experiment to calculate the fatigue threshold of the hydrogels [34,35]. Figure 3a shows the schematic diagram of the anti-fatigue of the PVA-AC conducting hydrogel. A small opening is cut above the hydrogel, and then the loading–unloading cycle experiment has been carried out in a water bath. The crack growth of the hydrogel is observed after several cycles. The crack propagation of the PVA-AC conductive hydrogel is caused by the destruction of the crystalline domain of the hydrogel [36,37]. The *λ_max_* value is finally determined by testing the crack propagation of the hydrogel under different strains. When *λ* is 1.7, the energy release rate of the PVA-AC conductive hydrogel is 40.89 J m^−2^, while, under the same test conditions, the energy release rate of pure PVA hydrogel is 18.37 J m^−2^ (Figure 3b). The fatigue threshold of the PVA-AC conductive hydrogel is approximately four times higher than that of the pure PVA hydrogel; the result demonstrates that AC increases the toughness and fatigue resistance of PVA hydrogels [38]. Figure 3c is a photo of the crack propagation of the PVA-AC conductive hydrogel after 30,000 rounds of cyclic loading when the *λ* is 1.7. The figure shows that after 30,000 loading cycles, the PVA-AC conductive hydrogel has not undergone significant crack growth, indicating that the PVA-AC conductive hydrogel has good fatigue resistance.

### 3.4. Sensing Performance of PVA-AC Conductive Hydrogel Strain Sensor

In order to further explore the application of the PVA-AC conductive hydrogel strain sensors, their sensing properties have been systematically investigated. The electrical signal is caused by a physical deformation that alters the path for electron transport, resulting in resistant changes. As the PVA-AC conductive hydrogel strain sensor is stretched, its conducting network is destroyed, and the tunneling channel between the conductive AC particles becomes longer, resulting in an increase in the resistance value. In order to analyze the strain sensitivity of the sensor, we calculate the gauge factor according to the relative resistance (Δ*R/R*_0_) under different strains. Figure 4a shows that the sensitivity value of our sensor can be divided into three regions in the strain range, and the corresponding gauge factors (*GF*) are 1.006 (0–50%), 1.667 (50–200%), and 2.193 (200–350%). Furthermore, due to the superior mechanical properties of the PVA-AC conducting hydrogel, we characterize the relative changes in the resistance signal of the sensor under different strains to prove the excellent electrical properties of our strain sensor. Figure 4b shows the variation in the relative resistance of the strain sensor at 50%, 100%, 150%, 200%, 250%, 300%, and 350% respectively, with 10 tensile release cycles per strain and essentially the same relative resistance per cycle. Moreover, our strain sensor still maintains a stable signal output in the range of a small strain and a medium strain, indicating that the sensor has superior tensile and electrical stability (Appendix A). The electrical signal response of the PVA-AC conducting hydrogel strain sensor not only changes in the tensile state, but the bending at different angles also causes the electrical signal and facilitates the widespread use of PVA-AC conductive hydrogels. Figure 4c shows the trend in the resistance values at 0°, 30°, 60°, and 90° states, respectively, while still recovering to the original resistance value from 90° to 0°, indicating that the strain sensor has excellent repeatability.

In order to further verify other excellent electrical performances of the sensor, we describe the linearity, hysteresis, and stability of the sensor in detail. Figure 4d demonstrates the linearity of relative resistance change and hysteresis at 0–350% strain, with results showing low hysteresis (<1.5%) and good linearity (0.982). Notably, the sensing ability and hysteresis display that the relative resistance values remain stable after cyclic loading at 100% strain for 10,000 cycles; the result indicates that the strain sensor has good repeatability and long-term stability (Figure 4f) [39,40]. The long-term stability of the PVA-AC conducting hydrogel strain sensor in the application of monitoring plant growth is longer than the previously reported cycling cycles, so the PVA-AC conducting hydrogel strain sensor is more suitable for long-term monitoring of plant growth. Thus, the low hysteresis and long-term stability of the PVA-AC hydrogel make it an ideal material for the preparation of strain sensors.

### 3.5. Long-Term Stable Plant Growth Monitoring

Due to the simple fabrication and easy integration of the PVA-AC hydrogel strain sensor, we were able to easily integrate our strain sensor on a stem of bamboo to verify its sensing performance. PVA-AC conductive hydrogel strain sensor with a three-layer structure facile integrated to verify its sensing performance by monitoring bamboo growth. VHB viscoelastic tape is used as the substrate and the encapsulation layer and the PVA-AC conductive hydrogel is used as the electrode. Conductive carbon cloth is connected at both ends of the electrode (Figure 5a). PVA-AC conductive hydrogel strain sensor is fixed on the stem of bamboo, the lead wire of the sensor is clamped by the fixture of the LCR meter, and the electrical signal is transmitted through the computer program, which can perform long-term stable plant growth monitoring (Figure 5b). Compared with rigid grating strain sensors, the soft and stretchable properties of the PVA-AC hydrogel and silicon-based elastomers allow our strain sensors to adapt well to bamboo growth.

The as-prepared sensor is used to monitor the growth of bamboo (Figure 5c). The growth of bamboo exerts an external pulling force on the PVA-AC conducting hydrogel strain sensor, resulting in an increase in the resistance. Figure 5c is a picture of the bamboo growth taken on the first day. During the long-term monitoring, after 14 days, we test the relative resistance of our sensor at the same time. Figure 5d shows the electrical response of the strain sensor to the continuous growth of bamboo for 14 days. During this period, the relative resistance ($\Delta R/R_{0}$) of the strain sensor increased by 1.15 kΩ, indicating that the sensor successfully monitors the growth of bamboo. Notably, we compute the growth status of the plant through the linear relationship of the resistance of the sensor under different strains. Figure 5e shows the growth length of bamboo during this period. We calculate the length of the bamboo increased by 28.199 mm according to the formula ($\Delta D=24\cdot \left(\Delta R/R_{0}\right)/0.982)$. The result indicates the feasibility of the sensor for plant growth monitoring. At the same time, the growth of plants can be clearly monitored in Figure 5f compared with that in Figure 5c. Therefore, we demonstrate that the low hysteresis, fatigue resistance, and stretchability of the PVA-AC conducting hydrogel strain sensors enable long-term stable monitoring of plant growth.

## 4. Conclusions

Overall, to address the challenge of measuring the long-term stability of plant growth, we successfully fabricate PVA-AC conductive hydrogel strain sensors by using more economical materials. Such sensors achieve excellent mechanical properties (including high stretchability, a low Young’s modulus, and low hysteresis), superior electrical signal response, and stable sensing performance. The results show that the strain sensor can timely obtain the growth status of the plant and stably measure the growth of the plant for 14 days. Using PVA-AC strain sensors to monitor plant growth may provide a solid basis for understanding plant growth regulation and promoting agricultural development. PVA-AC conductive hydrogel strain sensors not only provide an accurate platform for better monitoring plant growth but also offer a promising orientation for smart applications to improve the survival rate of plants in harsh environments by integrating with wireless devices. 

## Figures and Tables

**Figure 1 polymers-15-00090-f001:**
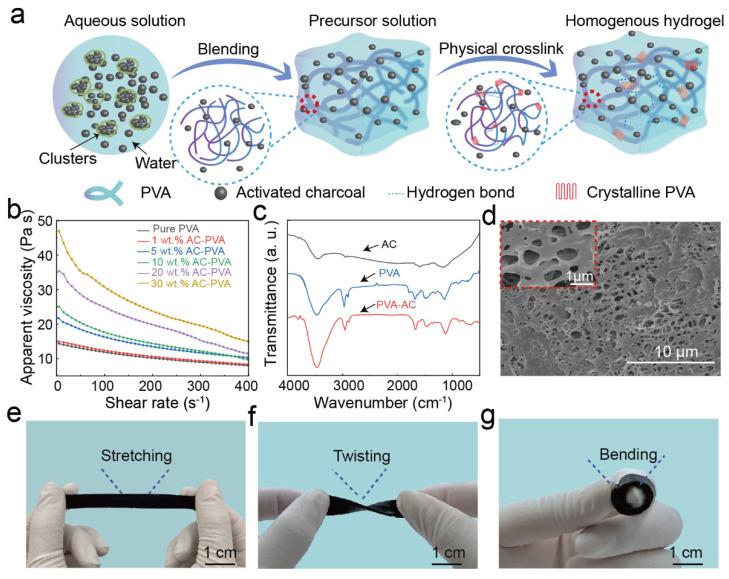
Design of PVA-AC conductive hydrogel. (**a**) Schematic diagram of the preparation of PVA-AC conductive hydrogel. (**b**) Apparent viscosity as a function of shear rate for PVA-AC inks of varying AC concentration. (**c**) FT-IR spectrum of AC, PVA, PVA-AC conductive hydrogel. (**d**) Scanning electron microscopy (SEM) images of PVA-AC conductive hydrogel cross-sectional morphologies. (**e**–**g**) Digital images of PVA-AC hydrogels against mechanical deformation: Stretching (**e**), twisting (**f**), and bending (**g**).

**Figure 2 polymers-15-00090-f002:**
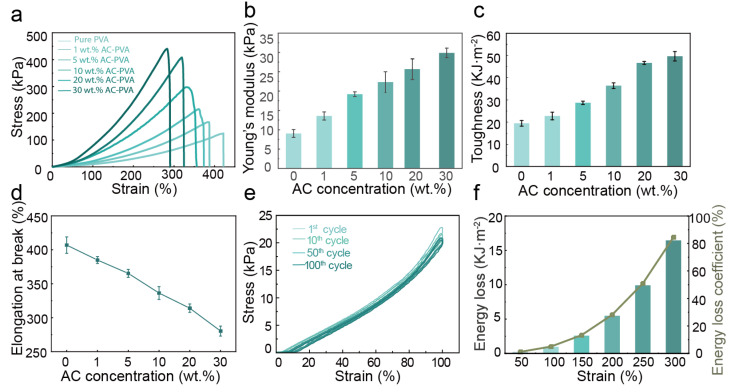
Mechanical properties of the PVA-AC conductive hydrogels. (**a**) Typical tensile stress–strain curves of PVA-AC conductive hydrogel. (**b**) Young’s modulus values of PVA-AC hydrogel with varying AC concentrations. (**c**) Toughness values of PVA hydrogel with varying AC concentrations. (**d**) Elongation at break of PVA hydrogel with varying AC concentrations. (**e**) Cyclic stress-strain curves of 5 wt.% PVA-AC conductive hydrogel. (**f**) Calculating dissipated energy and the energy loss coefficient of the PVA hydrogel with varying AC concentrations.

**Figure 3 polymers-15-00090-f003:**
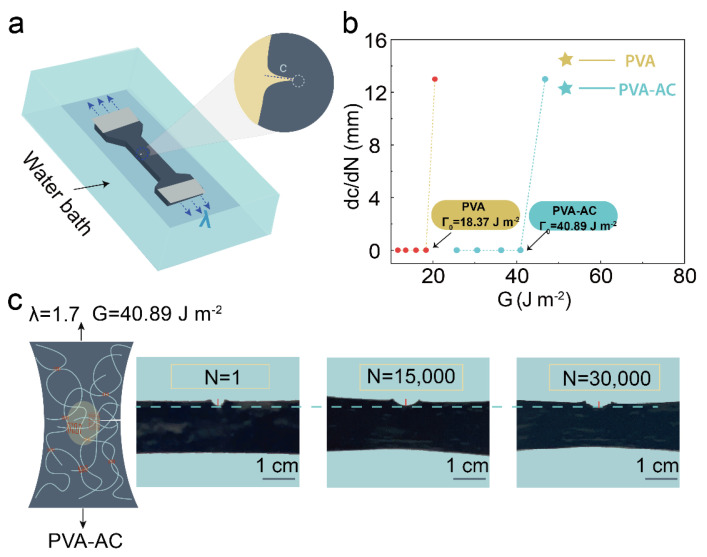
Fatigue-resistant design of PVA-AC hydrogels. (**a**) Single-notch cyclic loading test of PVA-AC conducting hydrogels (in the water bath). (**b**) Illustration of measuring crack extension per cycle *dc/dN* versus energy release rate *G* curves. Below this, the fatigue crack will not propagate under infinite cycles of loading. By definition, the fatigue threshold is equal to the critical energy release rate. (**c**) Photographs of the crack propagation of the PVA-AC conductive hydrogel at the cycle numbers of 1, 15,000, and 30,000 respectively.

**Figure 4 polymers-15-00090-f004:**
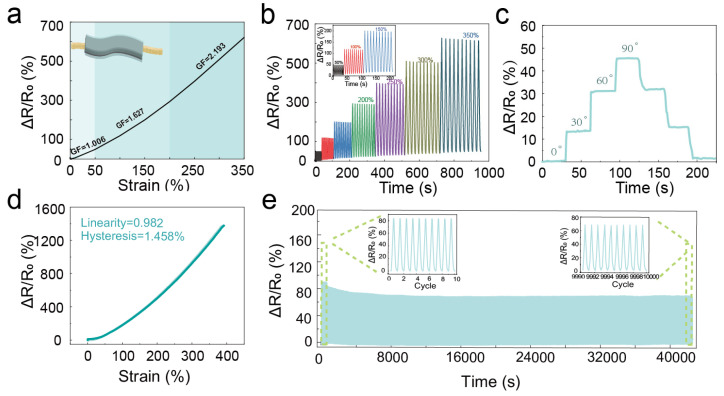
The sensing performance of 5 wt.% PVA-AC conducting hydrogel strain sensor. (**a**) Relative resistance changes as a function of the PVA-AC hydrogel strain sensor and the corresponding *GF*. (**b**) The relative resistance changes of the PVA-AC conducting hydrogel sensor at large strain. (**c**) Variation of the relative resistance of PVA-AC conducting hydrogel strain sensors at 0°, 30°, 60°, and 90°, respectively (**d**) Loading and unloading resistance responses of the PVA-AC hydrogel strain sensor with a strain of 300%, exhibiting low hysteresis (<1.5%) and linearity R^2^ = 0.98. (**e**) Relative resistance changes under cyclic stretching–releasing test for up to 10,000 cycles at 100% tensile strain.

**Figure 5 polymers-15-00090-f005:**
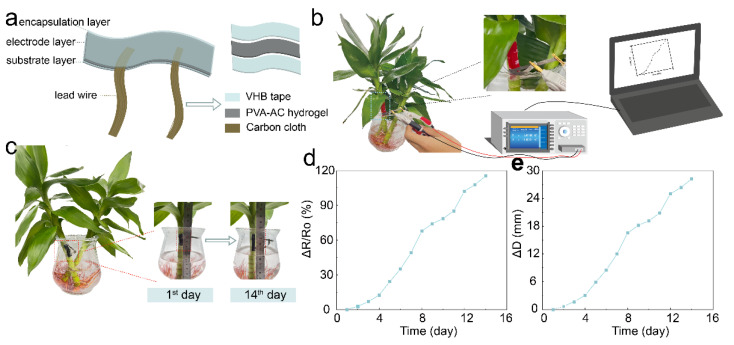
Design and application of strain sensor. (**a**) Schematic diagram of the fabrication of the strain sensor, including encapsulation layers, electrode layer, and lead wire. (**b**) For the monitoring of bamboo growth, the LCR tester is used to test the resistance value change of the strain sensor on the bamboo. (**c**) An image of the growth state of the bamboo on the first day and the fourteenth day when the sensor is attached to the bamboo. (**d**) Changes in resistance of sensors attached to bamboo during 14 days. (**e**) The growth curve of bamboo is calculated by the formula in 14 days.

## Data Availability

The raw/processed data required to reproduce these findings cannot be shared at this time due to technical limitations. They are available upon request.

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
