# Peer review of "Low Hysteresis and Fatigue-Resistant Polyvinyl Alcohol/Activated Charcoal Hydrogel Strain Sensor for Long-Term Stable Plant Growth Monitoring"

_polymers, 2022, doi:10.3390/polym15010090_

Round 1

Reviewer 1 Report

               Wang et al presents a novel route of monitoring stable plant growth via strain sensor. This strain sensor is fabricated from PVA/activated charcoal hydrogel and exhibit improved properties such as high stretch ability, low hysteresis losses, or high fatigue resistance. The presented strain sensor of the work overcome these innovative features and therefore useful for stable monitoring of agricultural products. The work presented is systematic and meaningful for the prospective application and worthy for publication in Polymers. However, some revision is required before acceptance. Few points are –

[1] Introduction is very preliminary. Instead, a lot of progress has been made on strain sensors which are based on soft polymer matrix with hardness below 65. Please refer 4-5 papers from MDPI to account different types of strain sensors with different stretch ability, hysteresis losses or fatigue resistance. In introduction, please state advantages or disadvantages or challenges or properties of use of PVA or AC or related materials which are useful for hydrogels based strain sensors. Some more recent studies can be referred such as –[a] https://doi.org/10.3390/polym13142322, [b] https://doi.org/10.1016/j.sna.2019.111712.

[2] In materials section, please provide physical properties of activated charcoal such as particle size, surface area, elemental composition in % or purity.

[3] In Figure 1a, second figure has typo, its “precursor solution” and NOT “precursor olution”. Please crosscheck the whole paper for possible typos.

[4] It will be good if authors provide SEM, Raman and XRD of activated charcoal characteristics in supporting information.

Good luck for revisions!

Author Response

Response to reviewers

At first, we deeply appreciate the editor and the reviewers for their hard work on our manuscript. From their comments, we learned a lot of new professional knowledge and experimental skills, etc. Below you will find our point-by-point response to the reviewers’ comments (Responses are marked in blue and sentences newly inserted into the revised manuscript are marked in red):

*********************************************************************

Response to Reviewer #1

Wang et al present a novel route of monitoring stable plant growth via strain sensor. This strain sensor is fabricated from PVA/activated charcoal hydrogel and exhibit improved properties such as high stretchability, low hysteresis losses, or high fatigue resistance. The presented strain sensor of the work overcome these innovative features and therefore useful for stable monitoring of agricultural products. The work presented is systematic and meaningful for the prospective application and worthy for publication in Polymers. However, some revision is required before acceptance. Few points are –

RESPONSE: We thank the reviewer for their recognition and constructive comments on our work. We have answered your questions in the response below.

  1. Introduction is very preliminary. Instead, a lot of progress has been made on strain sensors which are based on soft polymer matrix with hardness below 65. Please refer 4-5 papers from MDPI to account different types of strain sensors with different stretch ability, hysteresis losses or fatigue resistance. In introduction, please state advantages or disadvantages or challenges or properties of use of PVA or AC or related materials which are useful for hydrogels-based strain sensors. Some more recent studies can be referred such as [a] https://doi.org/10.3390/polym13142322, [b] https://doi.org/10.1016/j.sna.2019.111712.

RESPONSE: Thank you for giving us constructive suggestions. The reason we choose activated charcoal and PVA is that activated charcoal particles are insoluble and can aggregate or settle in an aqueous solution. PVA is a high molecular polymer, which becomes a viscous sol after dissolution and can effectively prevent the sedimentation of activated carbon particles. It is conducive to the uniform dispersion of activated charcoal particles in the PVA viscous solution to form a continuous and stable conductive network. According to the reviewer’s suggestion, we further clarify the properties of PVA in the revised introduction. Moreover, we have searched the literature on hydrogel strain sensors in recent years, and have added them in the introduction part accordingly as Refs. 15, 16 in the manuscript.

On Line 51: “We choose AC as the electrical phase and PVA as the mechanical phase to prepare conductive hydrogels. PVA is a high molecular polymer, which becomes a viscous solution after dissolution and can effectively prevent the sedimentation of AC particles. It is conducive to the uniform dispersion of AC particles in the PVA viscous solution to form a continuous and stable conductive network, which is beneficial for strain sensors to maintain stable properties.”

  1. Kumar, V.; Alam, M.; Manikkavel, A.; Song, M.; Lee, D.; Park, S. Silicone Rubber Composites Reinforced by Carbon Nanofillers and Their Hybrids for Various Applications: A Review. Polymers 2021, 13, 2322.
  2. Kumar, V.; Lee, G.; Singh, K.; Choi, J.; Lee, D. Structure-Property Relationship in Silicone Rubber Nanocomposites Reinforced with Carbon Nanomaterials for Sensors and Actuators. Sens. Actuators A Phys. 2020, 303, 111712.
  3. In materials section, please provide physical properties of activated charcoal such as particle size, surface area, elemental composition in % or purity.

RESPONSE: Regarding the physical properties of activated charcoal, we have modified the materials section of the manuscript as follows.

On Line 68: “activated charcoal (surface areas: 300 - 2,000 m2/g, particle size: ~ 1800 nm) was obtained from Nature’s Way Products (USA)”.

  1. In Figure 1a, second figure has typo, its “precursor solution” and NOT “precursor olution”. Please crosscheck the whole paper for possible typos.

RESPONSE: We are truly grateful for the reviewer’s meticulous check on our manuscript, and apologies for our inattentiveness at work. These problems have been modified in Figure 1 in the manuscript as shown below. At the same time, we have also checked the entire manuscript.

  1. It will be good if authors provide SEM, Raman and XRD of activated charcoal characteristics in supporting information.

RESPONSE: We are much obliged to the reviewer’s constructive comments. We totally agree with the reviewer on the point that SEM, Raman, and XRD imaging should be performed for the morphological investigation of our activated charcoal characteristics in supporting information. Unfortunately, due to the outbreak of COVID-19 in China, our university has been currently undergoing a complete campus lockdown and students have been requested to return home. All the public facilities are stopped using and are still far away from reopening. Therefore, we are terribly sorry to admit that SEM, Raman, and XRD imaging are not available within at least two months. In terms of the submission deadline, we have to submit the revision without adding these characterizations.

*********************************************************************

Once again, we acknowledge the reviewers’ comments and constructive suggestions very much, which are quite valuable in improving the quality of our original manuscript.

Reviewer 2 Report

Comments on the paper of Lina Wang, Zhilin Zhang, Jie Cao, Wenqian Zheng, Qi Zhao, Wenna Chen, Xinye Xu, Xiaoyu Luo, Qi Liu, Ximei Liu, Jingkun Xu, Baoyang Lu entitled “Low hysteresis and fatigue-resistant polyvinyl alcohol/activated charcoal hydrogel strain sensor for long-term stable plant growth monitoring”, Manuscript Number: polymers-2101365

Comments and Suggestions for Authors:

The manuscript deals with the subject of fabricate a conductive hydrogel strain sensor.

This sensor is obtained via through uniformly dispersing the conductive activated charcoal (AC) in high-viscosity  polyvinyl alcohol (PVA). The authors describe the influence of creation PVA-AC conductive hydrogel formed from crystalline domains of polyhydroxylated PVA cross-linked by hydrogen  bonding that provides sacrificial bonds for energy dissipation.

The study appears to be quite carefully conducted with a sound analysis. The results as well as their interpretation look well justified, and the paper is clearly written. I therefore generally recommend publication, however there are a couple of points that should first be addressed:

1)                 Why was polyvinyl alcohol used with such a high molecular weight and density?

2)                 Have you conducted tests using a different PVC and have you observed a change in the parameters of the obtained sensors?

3)                 Page 4, line 147“To ensure the stability of the PVA-AC conductive network, we further elaborate on its stability in terms of chemistry and microstructure.” - Sorry, but I don't understand how stability is further developed?

4)                  Has there been research into the pattern of plants other than bamboo?

5)                  Has a negative influence of the plant (humidity) on the change of sensor properties been observed?

Author Response

Response to reviewers

At first, we deeply appreciate the editor and the reviewers for their hard work on our manuscript. From their comments, we learned a lot of new professional knowledge and experimental skills, etc. Below you will find our point-by-point response to the reviewers’ comments (Responses are marked in blue and sentences newly inserted into the revised manuscript are marked in red):

*********************************************************************

Response to Reviewer #2

The manuscript deals with the subject of fabricating a conductive hydrogel strain sensor.

This sensor is obtained via through uniformly dispersing the conductive activated charcoal (AC) in high-viscosity polyvinyl alcohol (PVA). The authors describe the influence of the creation of PVA-AC conductive hydrogel formed from crystalline domains of polyhydroxylated PVA cross-linked by hydrogen bonding that provides sacrificial bonds for energy dissipation.

The study appears to be quite carefully conducted with sound analysis. The results as well as their interpretation look well justified, and the paper is clearly written. I therefore generally recommend publication, however, there are a couple of points that should first be addressed:

RESPONSE: We thank the reviewer for their recognition and constructive comments on our work. We have answered your questions in the response below.

  1. Why was polyvinyl alcohol used with such a high molecular weight and density? RESPONSE: Thanks to the reviewer for the question. The reason we choose PVA is that PVA is a high molecular polymer and its viscosity changes with different concentrations of PVA solutions. When the concentration of the PVA solution is too high, the activated charcoal particles will aggregate, which is not conducive to uniform dispersion, thus affecting the sensing performance of the sensor. At the same time, PVA provides a mechanical network for the sensor. When the PVA concentration is too low, it cannot meet the mechanical support of the sensor during the long-term growth of plants. Therefore, we choose the PVA solution with this concentration, which can achieve both excellent sensing performance and good mechanical properties.
  2. Have you conducted tests using a different PVC and have you observed a change in the parameters of the obtained sensors?

RESPONSE: Many thanks for the reviewer’s kind comments on our manuscript. In Figure S3, we tested the sensitivity value of the PVA-AC hydrogel strain sensor with different proportions under different strains and found that the sensitivity value of 5 wt.% PVA-AC hydrogel was the best. Meanwhile, sensitivity is an important parameter of sensor sensing performance to evaluate strain sensors, so the 5 wt.% PVA-AC hydrogel strain sensor was selected for anti-fatigue, sensing performances, and plant growth testing.

  1. Page 4, line 147“To ensure the stability of the PVA-AC conductive network, we further elaborate on its stability in terms of chemistry and microstructure.” - Sorry, but I don't understand how stability is further developed?

RESPONSE: We thank the reviewer for their suggestions. We demonstrate the stability of the PVA-AC conductive network by characterizing the PVA-AC conductive hydrogels with FT-IR spectroscopy and SEM. The FT-IR shows that the absorption peak of the conductive hydrogel hardly shifted after adding activated charcoal, which proves the stability of its chemical structure. At the same time, SEM characterization shows that the PVA-AC conductive hydrogel formes a dense network structure, and the activated charcoal particles are uniformly dispersed in the PVA hydrogel, forming a stable conductive network. We have also revised the relevant description as follows:

On Line 149: “To illustrate the stability of the PVA-AC conductive network, we further elaborate on its stability in terms of chemical structure and microstructure. The FT-IR shows that the absorption peak of the conductive hydrogel hardly shifted after adding activated charcoal, which proves the stability of its chemical structure (Figure 1c) [24–26].

  1. Has there been research into the pattern of plants other than bamboo?

RESPONSE: We are much obliged to the reviewer’s constructive comment. Besides the bamboo, we actually tried some other species like corn, rice, tomato and so on. But unfortunately our sensor shows certain mechanical constrain on these plants. So we didnot report the results here. On the other hand, one of our ongoning work is to make the whole sensor device softer and thinner in order to reduce the constrain and increase the overall sensing performance. These results will be reported elsewhere.

  1. Has a negative influence of the plant (humidity) on the change of sensor properties been observed?

RESPONSE: Thanks to the reviewer for the critical comment. In our work, we believe that the humidity exerts no negative effect on the performances of our sensor owing to the fact that a thin VHB tape with excellent sealing performance as the encapsulation layer ro prevent the hydration/dehydration of the PVA-AC conductive hydrogel. Correspondingly, the PVA-AC conductive hydrogel did not show obvious absorption or loss of water under different humidity conditions during the long term stability tests.

*********************************************************************

Once again, we acknowledge the reviewers’ comments and constructive suggestions very much, which are quite valuable in improving the quality of our original manuscript.
